# Mortality in association with antipsychotic medication use and clinical outcomes among geriatric psychiatry outpatients with COVID-19

Bienvenida Austria[1,2☯], Rehana Haque[1,2☯], Sukriti Mittal[1,2☯], Jamie Scott[1,2☯], Aninditha Vengassery[1,2☯], Daniel Maltz[3], Wentian Li[4], Blaine Greenwald[1,2], Yun Freudenberg-Hua [1,2,5] *

1 Division of Geriatric Psychiatry, Zucker Hillside Hospital, Glen Oaks, NY, United States of America, 2 Donald and Barbara Zucker School of Medicine at Hofstra/Northwell, Hempstead, NY, United States of America, 3 Information Services, Product Services and Management, Northwell Health, Lake Success, NY, United States of America, 4 Center for Genomics and Human Genetics, The Feinstein Institutes for Medical Research, Manhasset, NY, United States of America, 5 Litwin-Zucker Center for Alzheimer's Disease, The Feinstein Institutes for Medical Research, Manhasset, NY, United States of America

☯ These authors contributed equally to this work.

* yfreuden@northwell.edu

**Data Availability Statement:** The minimal anonymized data set necessary to replicate this

## Abstract

### Objectives

Older adults are particularly vulnerable to the negative consequences of antipsychotic exposure and are disproportionally affected by higher mortality from coronavirus disease 2019 (COVID-19). Our goal was to determine whether concurrent antipsychotic medication use was associated with increased COVID-19 mortality in older patients with preexisting behavioral health problems. We also report on findings from post-COVID follow-ups.

### Design

Retrospective observational study.

### Participants

Outpatients at a geriatric psychiatric clinic in New York City.

### Measurements

Demographic and clinical data including medication, diagnosis and Clinical Global Impression Severity (CGI-S) scales on outpatients who had COVID-19 between February 28th and October 1st 2020 were extracted from the electronic health records (EHR) from the hospital.

### Results

A total of 56 patients were diagnosed with COVID-19 (mean age 76 years; median age 75 years) and 13 (23.2%) died. We found an increased mortality risk for patients who were prescribed at least one antipsychotic medication at the time of COVID-19 infection (Fisher's exact test P = 0.009, OR = 11.1, 95% confidence interval: 1.4–96.0). This result remains significant after adjusting for age, gender, housing context and dementia (Logistic regression

study findings is uploaded as Supporting Information (S2 Table).

**Funding:** YFH received research grant from the National Institutes of Health, National Institute on Aging K08 AG054727. The funders had no role in study design, data collection and analysis, decision to publish, or preparation of the manuscript. The funder (National Institute on Aging) provided support in the form of salaries for the author [YFH], but did not have any additional role in the study design, data collection and analysis, decision to publish, or preparation of the manuscript. The specific roles of these authors are articulated in the 'author contributions' section.

**Competing interests:** No authors have competing interests.

P = 0.035, Beta = 2.4). Furthermore, we found that most patients who survived COVID-19 (88.4%) recovered to pre-COVID baseline in terms of psychiatric symptoms. Comparison of pre- and post-COVID assessments of CGI-S for 33 patients who recovered from COVID-19 were not significantly different.

## Conclusion

We observed a higher COVID-19 mortality associated with concurrent antipsychotics use in older patients receiving behavioral health services. The majority of patients in our geriatric clinic who recovered from COVID-19 appeared to return to their pre-COVID psychiatric function. More precise estimates of the risk associated with antipsychotic treatment in older patients with COVID-19 and other underlying factors will come from larger datasets and meta-analyses.

## Introduction

Long-term antipsychotic medication use has been unequivocally associated with increased mortality risk in older adults [1–3]. Among many adverse medical effects, antipsychotics have been reported to be associated with an increased risk for serious respiratory adverse events [4, 5] with a link to potential immune dysfunction [6]. The Coronavirus disease (COVID-19) caused by the SARS-CoV-2 virus infection disproportionally affects elderly with higher morbidity and mortality [7, 8]. The infection fatality rate was estimated to reach 4.9% for people aged 65–74 years and 14.2% for those 75 years and older [7]. In addition to older age, more risk factors have been identified, including but not limited to gender [9, 10], dementia, living at institutional care facilities, and medical comorbidities [11, 12]. Some of these risk factors may not be independent; for example, people with dementia are also more likely to live in care facilities. A population based observational study showed that living in care facilities is associated with an increased risk of COVID-19 mortality in comparison with living in independent housing, potentially through exposure to visitors and care workers, and due to poor underlying health [13].

Given that older adults are particularly vulnerable to the negative consequence of antipsychotic exposure as well as to COVID-19, we evaluated clinical data from a community-based geriatric psychiatric outpatient clinic in New York City (NYC) to determine whether long-term use of antipsychotics in the context of COVID-19 would increase the risk of mortality. Moreover, neurological and psychiatric complications are common during the acute phase of COVID-19, including delirium [14], stroke, intracerebral hemorrhage, CNS vasculitis, encephalopathy, new-onset psychosis, neurocognitive syndrome and affective symptoms [15, 16]. An increased incidence of psychiatric disorders has been reported in early post-COVID follow-ups [17, 18] and psychiatric patients may require additional mental health support during the pandemic [18, 19]. Therefore, we wonder whether older patients with preexisting mental illness show more post-COVID psychiatric consequences. We took advantage of our out-patient follow-ups to compare pre- and post COVID-19 overall clinical psychiatric symptom severity as a secondary outcome.

## Methods

This study is a single-center retrospective observational study and approved by the Institutional Review Board of Northwell Health (# 20–0200). The outpatients and their caregivers

reported COVID-19 diagnosis and related symptoms to the treating psychiatrists at the Zucker Hillside Hospital (ZHH) of the Northwell Health system. We included all patients who had COVID-19 between February 28th and October 1st 2020. The COVID-19 diagnosis was confirmed with reverse transcription polymerase chain reaction (RT-PCR) during the symptomatic phase or confirmed using serology test after the suspected acute COVID-19 infection. The outcome is defined as whether the patient expired during the acute COVID-19 hospitalization. Clinical data including the use of antipsychotics at the time of COVID-19 (exposure variable), antidepressants, age, gender, dementia diagnosis (defined as ICD-10 codes starting with F01, F02, F03, F04, F09, G30, G31, and G32) as well as Clinical Global Impression severity scales (CGI-S) [20] were extracted from hospital EHR and manually quality checked. The CGI-S, which is integrated in the EHR but not as a required field, is routinely used by the treating psychiatrists during clinical assessments. CGI-S scales from September 1st 2019 to the date of COVID-19 for each patient are defined as pre-COVID (T0) and from the date of COVID-19 till October 22nd 2020 (time point of EHR data extraction) are defined as post-COVID (T1) measures. Living arrangement, estimated timing of COVID-19 infection, and whether the patients survived were obtained through chart review. We estimated the date of COVID-19 infection using the date of the reported symptom onset by patients and their caregivers or the date of positive results of the RT-PCR test. In addition to CGI-S, the treating psychiatrists (who typically follow the same patients longitudinally) were surveyed in October 2020 for their impression of whether the patients recovered to their pre-COVID functional baseline. The acute symptoms of COVID-19 were treated at various health facilities including urgent care centers, primary care physicians' and medical specialists' offices and inpatient medical facilities. Following the Strengthening the Reporting of Observational Studies in Epidemiology (STROBE) statement [21], a check list is presented as **S1 Table**.

Statistical analysis was performed using R version 3.5.0 [22]. Given the limited sample sizes we use Fisher's exact test to test for univariate association between binary variables such as patient survival with antipsychotic use [23]. The odds ratio (OR) and 95% confidence interval are calculated using standard methods [24]. We used multivariable logistic regression to account for potential confounding factors including age, gender, living arrangement, and dementia diagnosis. As a sensitivity analysis, antidepressant use was added to the model. Shapiro-Wilk normality test indicated non-normal distribution of differences between CGI-S at T0 and T1 (P = 0.001). Accordingly, we used both paired t-test to compare the means and Wilcoxon paired signed rank test to compare the medians of individual pre- (T0) and post-COVID-19 (T1) CGI-S.

## Results

A total of 56 patients (mean age 76.0 ± 8.5 years; median age: 74.5 years; interquartile range: 13.0 years) (**Table 1**; **S2 Table**) with confirmed COVID-19 were reported to the treating psychiatrists between February and September 2020. Thirteen (23.2%) patients died while being hospitalized for confirmed COVID-19. In this cohort, gender is not significantly associated with mortality (Fisher's exact test P = 0.31) but higher age is associated with higher risk of death (P = 0.05; Beta = 0.08), which is consistent with previously reported findings [7]. In our cohort, female patients had a higher mean age than male patients (77.8 vs. 71.5 years old) (t-test P = 0.01). Institutional living arrangement (Fisher's exact test P = 0.10) and dementia diagnosis (P = 0.16) are not significantly associated with COVID-19 mortality in this cohort.

We observed a significant association between being prescribed an antipsychotic medication at the time of COVID-19 and an increased mortality risk (Fisher's exact test P = 0.009, OR = 11.1, 95% confidence interval: 1.4–96.0) (**Fig 1**). The association between antipsychotics

**Table 1. Patient characteristics stratified by outcome of COVID-19 patients.**

|  |  | Survived | Expired | OR (95% CI) | P |
|---|---|---|---|---|---|
| N (Total = 56) |  | 43 | 13 |  |  |
| age_covid (mean (SD)) |  | 74.72 (8.04) | 80.15 (9.15) | 1.1 (1.0–1.2) | 0.05 |
| gender (%) | F | 29 (67.4) | 11 (84.6) | 2.6 (0.5–27.5) | 0.31 |
|  | M | 14 (32.6) | 2 (15.4) |  |  |
| Institutional living (%) | Yes | 6 (14.0) | 5 (38.5) | 3.7 (0.7–19.4) | 0.10 |
|  | No | 37 (86.0) | 8 (61.5) |  |  |
| Dementia (%) | Yes | 10 (23.3) | 6 (46.2) | 2.8 (0.6–12.4) | 0.16 |
|  | No | 33 (76.7) | 7 (53.8) |  |  |
| Antipsychtics (%) | Yes | 22 (51.2) | 12 (92.3) | 11.1 (1.4–96.0) | 0.009 |
|  | No | 21 (48.8) | 1 (7.7) |  |  |

The association between survival and age was tested by logistic regression, whereas to test the associations of binary variables, Fisher's exact test was used. All P-values are unadjusted for covariables. Institutional Living is defined as living at an assisted living facility or at a rehabilitation facility. Antipsychotics: patients who were prescribed at least one antipsychotic medication at the time of COVID-19 infection to treat preexisting behavioral health disorders.

and COVID-19 mortality remained significant (P = 0.035, Beta = 2.4) after adjusting for age, gender, living arrangements (home vs. institutions), and dementia diagnosis. Further adjusting for antidepressants use as a sensitivity analysis didn't change the association (S3 Table). In univariable analysis, antipsychotics use was not significantly associated with age, gender, living

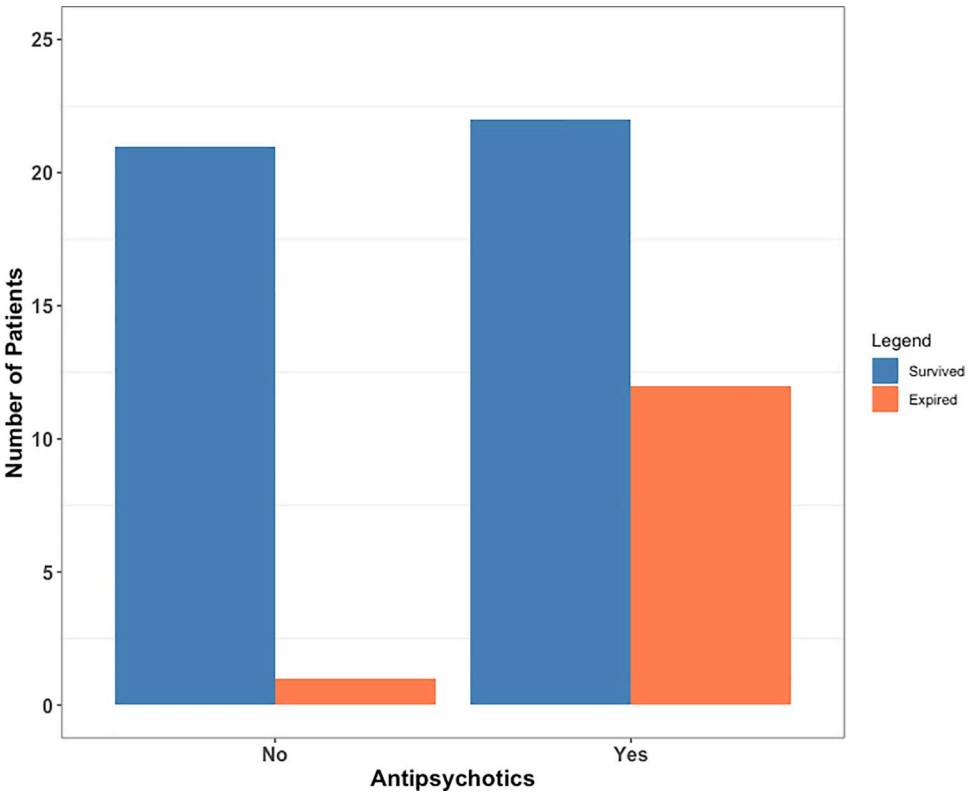

**Fig 1. COVID-19 mortality of patients stratified by antipsychotics use.** Number of geriatric psychiatric patients who survived and died from COVID-19. X-axis: "No": not being prescribed an antipsychotic medication (steel blue); "Yes": being prescribed an antipsychotic medication before COVID-19 (coral).

arrangements or dementia (S4 Table). Among the 34 patients treated with antipsychotics, only two (5.9%) received monotherapy with a typical antipsychotic medication (haloperidol) and both recovered. Another two patients were prescribed a combination of a typical and an atypical antipsychotic medication and both expired. The remaining 30 patients were all taking atypical antipsychotics and 10 of them (33.3%) expired. The typical antipsychotics used are haloperidol, paliperidone, fluphenazine, trifluoperazine and the atypical antipsychotics include olanzapine, aripiprazole, quetiapine, risperidone, and clozapine. These antipsychotics were prescribed for a variety of conditions including dementia with behavior disorders, bipolar disorder, schizophrenia, schizoaffective disorder, psychosis, and major depressive disorder. Other psychotropic medications include mood stabilizers such as lamotrigine and valproic acid, gabapentin, anxiolytics such as benzodiazepines or buspirone and cholinesterase inhibitors such as donepezil.

The out-patient setting, in which patients being followed up by the same psychiatrist, further presents an opportunity to investigate how elderly patients with behavioral health conditions who survived COVID-19 recovered longitudinally in terms of psychiatric function. When surveyed, the treating geriatric psychiatrists reported that the majority of the patients—38 out of 43 patients (88.4%)—recovered to pre-COVID level of functioning. In addition, 33 patients had at least one CGI-S [20] score—a structured rating on how mentally ill a patient presents—for both pre- and post-COVID visits. Comparing the aggregated T0 and T1 CGI-S scores, we found no significant difference (paired t-test for differences of the mean CGI-S: $P = 0.23$ and Wilcoxon paired signed rank test for median: $P = 0.35$) (Fig 2).

## Discussion

As older adults are particularly vulnerable to adverse consequence of antipsychotic exposure [1–3] and have higher mortality risk from COVID-19 [7, 8, 25], we evaluated EHR data from a community-based geriatric psychiatry clinic during the height of the COVID-19 pandemic in NYC. We investigated whether concurrent antipsychotics prescription is associated with increased risk of COVID-19 mortality in this geriatric population receiving behavioral health services. We found that concurrent antipsychotic exposure was associated with an increased mortality risk in context of acute COVID-19. This finding remains significant after controlling for age, gender, living arrangement, and the presence of dementia.

A previous report showed that patients with a pre-existing psychiatric diagnosis while hospitalized for COVID-19 had a mortality rate that is twice as high compared to those without a psychiatric condition [26]. The authors speculated that there may be a causal relationship between psychiatric disorders and/or psychotropic medication and the immune system. A systematic screening of protein-protein interactions to identify critical factors for SARS-CoV-2 infection found that typical antipsychotics may have antiviral effects against COVID-19 while atypical antipsychotics do not [27]. As a proof of concept, the authors then utilized medical billing data to confirm their drug repurposing targets and found that half as many of the new users of typical antipsychotics (N = 13) compared with the new users of atypical antipsychotics (N = 26) progressed to the point of requiring mechanical ventilation. Another observational study found a reduced risk for intubation or death from COVID-19 to be associated with antidepressants as home medications, if they were continued within 48 hours of hospital admission [28]. A significant association of antipsychotics use with intubation or death (Hazard ration = 1.98) was observed in the crude analysis, but not in the multivariate analysis of this earlier study. These findings underscore both the importance and complexity of investigating psychotropic medication use and COVID-19 outcomes.

We found that despite mental status changes being a common symptom in COVID-19, geriatric behavioral health patients in our study did not show worsening of psychiatric

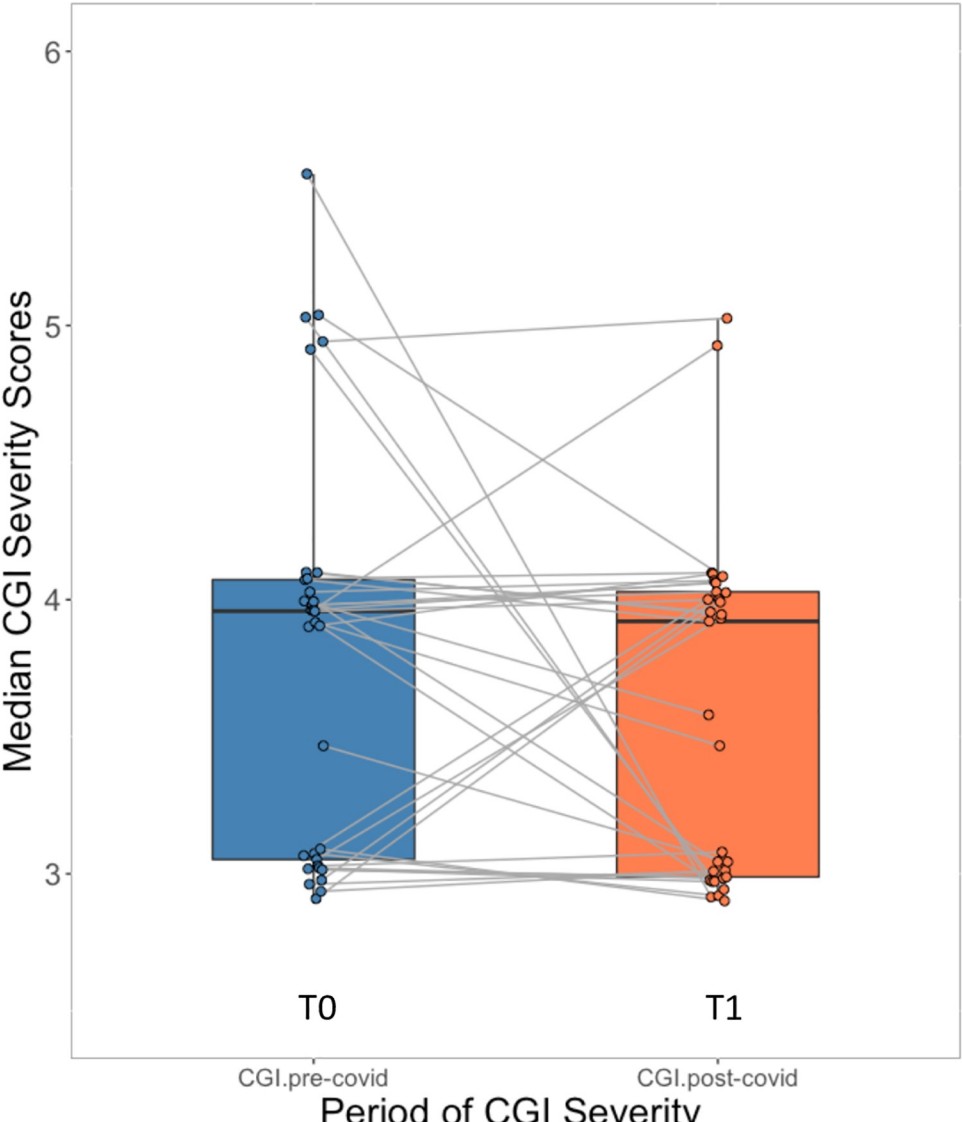

**Fig 2. Median CGI-S scores before and after COVID-19.** "CGI.pre-covid" (T0): Median Clinical Global Impression severity scales (CGI-S) before COVID-19 (steel blue); "CGI.post-covid" (T1): Median CGI-S during follow-up after COVID-19 (coral). Each point represents the median CGI-S score of one patient and the gray lines connect the pre- and post-COVID CGI-S for each patient. Thirty-three patients have both pre- and post-COVID CGI-S scores (Paired t-test for mean P = 0.23, Wilcoxon paired signed rank test for median: P = 0.35).

function during the short-term post-COVID period. There had been considerations that COVID-19 may potentially trigger future neurodegeneration through neuroinflammatory mechanisms [29]. One large study that included adults at all ages indicated a bidirectional interplay between COVID-19 and psychiatric problems [17]. They found that survivors of COVID-19 appear to be at increased risk of psychiatric sequelae including dementia, and a psychiatric diagnosis might be an independent risk factor for negative outcomes of COVID-19. A potential mechanism may include damage to blood-brain barrier mediated by a massive increase in circulating pro-inflammatory factors [30]. It is critical that COVID-19 survivors receive longitudinal follow-up neuropsychiatric assessments to monitor and investigate the long-term neurocognitive outcome.

Our study has several limitations. First, the odds ratio for the association of antipsychotics and mortality had wide confidence interval that is likely due to the small sample size. Second, adjusting for covariables in a small sample may lead to overfitting of the multiple logistic regression model. Third, mortality could be affected by additional demographic and clinical factors [31] that were not accounted for, which further limits the generalizability of our findings. Despite the sample size limitation, our exploratory results nevertheless present an addition to the discussion on the interplay between COVID-19 and mental health [18] as well as the post-COVID recovery [32]. Given the retrospective observational character of our study, we emphasize that although we report a significant correlation between long-term antipsychotic medication usage and COVID-19 mortality, any hypothetical causal relation remains to be understood. Antipsychotics are frequently prescribed in geriatric patients to treat a wide spectrum of disorders. Our study is statistically underpowered to investigate the effects of subtypes of antipsychotic medication as well as the combined effects of different psychotropics and underlying behavioral health conditions.

## Conclusions

We observed a higher COVID-19 mortality associated with antipsychotics use in older patients receiving behavioral health services. The majority of COVID-19 survivors recovered without overt psychiatric sequelae during short-term follow-ups. Future studies with larger data sets and meta-analysis will provide more precise estimates of the degree of risk of antipsychotic treatment in the elderly with comorbid COVID-19 so that providers and consumers can consider risks, benefits, and alternatives.

## Supporting information

**S1 Table. STROBE checklist.**
(DOCX)

**S2 Table. Deidentified characteristics of patients.** "0" ="Yes, "1" = No. For the purpose of deidentification, we included age range of the patients. Institutional Living is defined as living at an assisted living facility or at a rehabilitation facility. Antipsychotics: patients who were prescribed at least one antipsychotic medication at the time of COVID-19 infection to treat preexisting behavioral health disorders. Antidepressants: patients who were prescribed at least one antipsychotic medication at the time of COVID-19 infection. CGI-S: Clinical Global Impression severity scales. T0_CGI:_CGI-S scales from September 1st 2019 to the date of COVID-19 infection (pre-COVID); T1_CGI: CGI-S scales from the date of COVID-19 till October 22nd 2020 (post-COVID). SD: standard deviation. IQR .25: the 25% bound of Interquartile range. IQR .75: the 75% bound of Interquartile range.
(XLSX)

**S3 Table. Mortality in patients with and without concurrent antipsychotics use.** Multivariable logistic regression adjusting for age, gender, antidepressant use, living arrangement and dementia diagnosis.
(XLSX)

**S4 Table. Patient characteristics stratified by antipsychotics use of COVID-19 patients.** Antipsychotics: patients who were prescribed at least one antipsychotic medication at the time of COVID-19 infection to treat preexisting behavioral health disorders. Institutional Living is defined as living at an assisted living facility or at a rehabilitation facility. For the association between antipsychotics and age, logistic regression was used. For the associations of binary

variables, Fisher's exact test was used. P-values are unadjusted for covariates.
(XLSX)

## Author Contributions

**Conceptualization:** Bienvenida Austria, Rehana Haque, Sukriti Mittal, Jamie Scott, Aninditha Vengassery, Blaine Greenwald, Yun Freudenberg-Hua.

**Data curation:** Bienvenida Austria, Rehana Haque, Sukriti Mittal, Jamie Scott, Aninditha Vengassery, Daniel Maltz, Yun Freudenberg-Hua.

**Formal analysis:** Wentian Li, Yun Freudenberg-Hua.

**Investigation:** Bienvenida Austria, Rehana Haque, Sukriti Mittal, Jamie Scott, Aninditha Vengassery, Yun Freudenberg-Hua.

**Methodology:** Wentian Li, Yun Freudenberg-Hua.

**Resources:** Daniel Maltz, Yun Freudenberg-Hua.

**Supervision:** Wentian Li, Blaine Greenwald.

**Visualization:** Yun Freudenberg-Hua.

**Writing – original draft:** Rehana Haque, Yun Freudenberg-Hua.

**Writing – review & editing:** Bienvenida Austria, Rehana Haque, Sukriti Mittal, Jamie Scott, Aninditha Vengassery, Wentian Li, Blaine Greenwald, Yun Freudenberg-Hua.

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
