## [Decision Letter · Decision Letter 0]

6 Jul 2021

PONE-D-21-17894

Acute and Post-acute Outcome of Geriatric Psychiatry Outpatients with Coronavirus Disease (COVID-19)

PLOS ONE

Dear Dr. Freudenberg-Hua,

Thank you for submitting your manuscript to PLOS ONE. After careful consideration, we feel that it has merit but does not fully meet PLOS ONE’s publication criteria as it currently stands. Therefore, we invite you to submit a revised version of the manuscript that addresses the points raised during the review process.

We look forward to receiving your revised manuscript.

Kind regards,

Ismaeel Yunusa, PharmD, PhD

Academic Editor

PLOS ONE

Journal Requirements:

"YFH received research grant from the National Institutes of Health, National Institute on Aging K08 AG054727.

We note that one or more of the authors is affiliated with the funding organization, indicating the funder may have had some role in the design, data collection, analysis or preparation of your manuscript for publication; in other words, the funder played an indirect role through the participation of the co-authors. If the funding organization did not play a role in the study design, data collection and analysis, decision to publish, or preparation of the manuscript and only provided financial support in the form of authors' salaries and/or research materials, please do the following:

a. Review your statements relating to the author contributions, and ensure you have specifically and accurately indicated the role(s) that these authors had in your study. These amendments should be made in the online form.

b. Confirm in your cover letter that you agree with the following statement, and we will change the online submission form on your behalf: 

“The funder provided support in the form of salaries for authors [insert relevant initials], but did not have any additional role in the study design, data collection and analysis, decision to publish, or preparation of the manuscript. The specific roles of these authors are articulated in the ‘author contributions’ section.

Additional Editor Comments (if provided):

Please ensure you follow the STROBE (See: https://www.equator-network.org/reporting-guidelines/strobe/) recommendation in the reporting of your revised manuscript while addressing comments from the reviewers. Resubmit the revised manuscript along with a completed STROBE checklist.

Reviewers' comments:

Reviewer's Responses to Questions

**Comments to the Author**

1. Is the manuscript technically sound, and do the data support the conclusions?

Reviewer #1: Partly

Reviewer #2: Partly

2. Has the statistical analysis been performed appropriately and rigorously? 

Reviewer #1: Yes

Reviewer #2: N/A

3. Have the authors made all data underlying the findings in their manuscript fully available?

Reviewer #1: Yes

Reviewer #2: No

4. Is the manuscript presented in an intelligible fashion and written in standard English?

Reviewer #1: Yes

Reviewer #2: Yes

5. Review Comments to the Author

Reviewer #1: This is a retrospective observational study in a community-based geriatric psychiatric clinic. The authors found that concurrent antipsychotic use was associated with increased risk of mortality from acute COVID-19 infection in older patients with preexisting behavioral health problems, even after adjusting for age, gender, living arrangement, and dementia. They also compared pre- and post-COVID CGI-S scores of the patients, and found that 88% of the patients recovered to pre-COVID level of functioning. The authors did a great job interpreting their results and their discussion was very interesting and important.

• Major comments –

o The odds ratio of mortality for patients using antipsychotics has a wide confidence interval, probably due to the small sample size. Furthermore, of the 56 people, 13 expired. Using the 10:1 event:variable rule of thumb to inspect potential overfitting of logistic regression, the authors should be very careful when interpreting their results.

o The primary outcome is COVID-19 mortality. However, older adults are susceptible to other types of mortality. Were people counted as expired from COVID if their main cause of death on their death certificate was COVID? Please address in more detail how the authors handled competing risks.

o The study’s secondary outcome: pre- and post- covid CGI-S was reported by the patient’s treating geriatric psychiatrists. In line 106, the authors mentioned CGI-S is extracted from EHR. But in line 165: “when surveyed, the treating geriatric psychiatrists reported …”. Please specify if CGI-S is routinely evaluated and extracted from EHR, or how the authors reduced the potential bias from outcome reporting.

o The authors can consider a table that summarizes study characteristics of patients with and without antipsychotics use, because the clinical characteristics of these two populations may be very different.

• Minor comments–

o Line 143: The authors mentioned they adjusted for age, gender, living arrangements, and dementia diagnosis in their logistic regression, but given the excel supplementary file, should also add “adjusted for antidepressants”?

o Line 144-149: Can consider summarizing a table of mortality stratified by typical, atypical, and combination antipsychotics.

o Line 154: Not specified if psychotropic agents use were counted as antipsychotics use. If not, should be more explicit.

Reviewer #2: TITLE

I suggest a more focused title. “Acute and Post-acute Outcome” is too general. What do you mean for Acute and Post-acute Outcome? Please clarify

INTRODUCTION.

The present manuscript provides a very constructive background in terms of side effects of antipsychotic medication and COVID-19. However, I suggest a more specific review of the literature on the importance of psychological support in improving COVID-19 complications.

- Accordingly, you might cite:

1) Hossain MM, Tasnim S, Sultana A, Faizah F, Mazumder H, Zou L, McKyer ELJ, Ahmed HU, Ma P. Epidemiology of mental health problems in COVID-19: a review. F1000Res. 2020 Jun 23;9:636. doi: 10.12688/f1000research.24457.1.

2) Demeco A, Marotta N, Barletta M, et al. Rehabilitation of patients post-COVID-19 infection: a literature review. J Int Med Res. 2020;48(8):300060520948382. doi:10.1177/0300060520948382

-“The Coronavirus disease (COVID-19) caused by the SARS-CoV-2 virus infection disproportionally affects elderly with higher morbidity and mortality.” Please clarify this period and/or provide a reference

METHODS.

-Did you calculate the sample size?

-Did you run a normality test to justify the use of paired sample t-test?

- “pre- and post-COVID-19 CGI-S for patients with available data”. Please state the timing making explicit T0 and T1.

- “CGI-S scales from September 1st 2019 to the date of COVID-19 for each patient are defined as pre-COVID and from the date of COVID-19 till October 22nd 2020 (time point of EHR data extraction) are defined as post-COVID measures”. Could the interval of time between COVID-19 diagnosis and CGI-S scale affect the results?

RESULTS.

Please report data as mean and standard deviation or median and interquartile range.

DISCUSSION.

“As older adults are particularly vulnerable to the negative consequence of antipsychotic exposure and to higher mortality risk from COVID-19”. Please add a reference.

6. PLOS authors have the option to publish the peer review history of their article (what does this mean?). If published, this will include your full peer review and any attached files.

Reviewer #1: No

Reviewer #2: No

---

## [Author Response · Author response to Decision Letter 0]

9 Aug 2021

Reviewer #1: This is a retrospective observational study in a community-based geriatric psychiatric clinic. The authors found that concurrent antipsychotic use was associated with increased risk of mortality from acute COVID-19 infection in older patients with preexisting behavioral health problems, even after adjusting for age, gender, living arrangement, and dementia. They also compared pre- and post-COVID CGI-S scores of the patients, and found that 88% of the patients recovered to pre-COVID level of functioning. The authors did a great job interpreting their results and their discussion was very interesting and important.

• Major comments –

1. The odds ratio of mortality for patients using antipsychotics has a wide confidence interval, probably due to the small sample size. Furthermore, of the 56 people, 13 expired. Using the 10:1 event:variable rule of thumb to inspect potential overfitting of logistic regression, the authors should be very careful when interpreting their results.

> We agree with reviewer #1 that the wide OR is due to the small sample size and that we should be careful to interpret the result. We have further emphasized that point in the Discussion (page 12). We wrote: “First, the odds ratio for the association of antipsychotics and mortality had wide confidence interval that is likely due to the small sample size. Second, adjusting for covariables in a small sample may lead to overfitting of the multiple logistic regression model.”

2. The primary outcome is COVID-19 mortality. However, older adults are susceptible to other types of mortality. Were people counted as expired from COVID if their main cause of death on their death certificate was COVID? Please address in more detail how the authors handled competing risks.

> This is a valid statement. We didn’t obtain death certificate. The patients who died while hospitalized for COVID-19 were considered as COVID-19 mortality. However, it is well known that comorbidities contribute to COVID-19 mortality. In Discussion section on page 12 we added “Third, mortality could be affected by additional demographic and clinical factors(30) that were not accounted for, which further limits the generalizability of our findings.”

3. The study’s secondary outcome: pre- and post- covid CGI-S was reported by the patient’s treating geriatric psychiatrists. In line 106, the authors mentioned CGI-S is extracted from EHR. But in line 165: “when surveyed, the treating geriatric psychiatrists reported …”. Please specify if CGI-S is routinely evaluated and extracted from EHR, or how the authors reduced the potential bias from outcome reporting.

> We thank the reviewer for pointing out the need to clarify. We made clarifications in the Methods section on page 5-6 “The CGI-S, which is integrated in the EHR but not as a required field, is routinely used by the treating psychiatrists during clinical assessments” and on page 6 “In addition to CGI-S, the treating psychiatrists (who typically follow the same patients longitudinally) were surveyed in October 2020 for their impression of whether the patients recovered to their pre-COVID functional baseline.” All patients included in this study were under the active psychiatric care of the study authors.

4. The authors can consider a table that summarizes study characteristics of patients with and without antipsychotics use, because the clinical characteristics of these two populations may be very different.

> As requested by reviewer #1 we made an additional supporting Table S4 “Patient Characteristics stratified by antipsychotics use”. We added the sentence on page 8 “In univariable analysis, antipsychotics use was not significantly associated with age, gender, living arrangements or dementia (Table S4).”

• Minor comments–

o Line 143: The authors mentioned they adjusted for age, gender, living arrangements, and dementia diagnosis in their logistic regression, but given the excel supplementary file, should also add “adjusted for antidepressants”?

> We thank the reviewer #1 for the suggestions. As the reviewer mentioned in the Major comments, the sample size is too small to allow adjusting for many covariables. We did not adjust for antidepressants as a variable in our main model. In a larger study, all medications should be adjusted. We did a sensitivity analysis that included antidepressants and we wrote on page 8 “Further adjusting for antidepressants use as a sensitivity analysis didn’t change the association (Table S3)”. We included antidepressants in the supplementary table S2 thinking that other researchers who conduct meta-analysis might find this information helpful. 

o Line 144-149: Can consider summarizing a table of mortality stratified by typical, atypical, and combination antipsychotics.

> As indicated on page 9 line 294, only two patients received typical antipsychotics as monotherapy. Given the extremely small sample size we don’t think it’s meaningful to present a stratified table. The information is included in the Table S2 for those who are interested.

o Line 154: Not specified if psychotropic agents use were counted as antipsychotics use. If not, should be more explicit.

> We did not analyze “other psychotropics” in this study and thus, we removed this category.

Reviewer #2: TITLE

I suggest a more focused title. “Acute and Post-acute Outcome” is too general. What do you mean for Acute and Post-acute Outcome? Please clarify

> We thank reviewer #2 for this constructive suggestion and we changed the title to “Mortality and psychiatric outcome among geriatric psychiatry outpatients with Coronavirus disease (COVID-19)”.

INTRODUCTION.

The present manuscript provides a very constructive background in terms of side effects of antipsychotic medication and COVID-19. However, I suggest a more specific review of the literature on the importance of psychological support in improving COVID-19 complications.

- Accordingly, you might cite:

1) Hossain MM, Tasnim S, Sultana A, Faizah F, Mazumder H, Zou L, McKyer ELJ, Ahmed HU, Ma P. Epidemiology of mental health problems in COVID-19: a review. F1000Res. 2020 Jun 23;9:636. doi: 10.12688/f1000research.24457.1.

2) Demeco A, Marotta N, Barletta M, et al. Rehabilitation of patients post-COVID-19 infection: a literature review. J Int Med Res. 2020;48(8):300060520948382. doi:10.1177/0300060520948382

> We thank reviewer #2 for pointing to these references and we cited Hossain et al paper on page 5 and we cited both papers on page 12. 

-“The Coronavirus disease (COVID-19) caused by the SARS-CoV-2 virus infection disproportionally affects elderly with higher morbidity and mortality.” Please clarify this period and/or provide a reference

> We added citations to this sentence (page 4 line 119).

METHODS.

-Did you calculate the sample size?

> We did not calculate the sample size, as this is a retrospective cohort study and on page 5 line 149 we wrote: “We included all patients who had COVID-19 between February 28th and October 1st 2020” and in Results section on page 7 “A total of 56 patients…”.

-Did you run a normality test to justify the use of paired sample t-test?

 > We thank reviewer #2 for this suggestion. We updated the Methods section on page 6-7: “Shapiro-Wilk normality test indicated non-normal distribution of differences between CGI-S at T0 and T1 (P=0.001). Accordingly, we used both paired t-test to compare the means and Wilcoxon paired signed rank test to compare the medians of individual pre- (T0) and post-COVID-19 (T1) CGI-S.” We updated the results on page 10: “Comparing the aggregated T0 and T1 CGI-S scores, we found no significant difference (paired t-test for differences of the mean CGI-S: P=0.23 and Wilcoxon paired signed rank test for median: P=0.35) (Fig 2).”

Since these tests are not significant, we did not explore the usage of other tests.

- “pre- and post-COVID-19 CGI-S for patients with available data”. Please state the timing making explicit T0 and T1.

> We introduced the term T0 and T1 on page 6: “CGI-S scales from September 1st 2019 to the date of COVID-19 for each patient are defined as pre-COVID (T0) and from the date of COVID-19 till October 22nd 2020 (time point of EHR data extraction) are defined as post-COVID (T1) measures.”

- “CGI-S scales from September 1st 2019 to the date of COVID-19 for each patient are defined as pre-COVID and from the date of COVID-19 till October 22nd 2020 (time point of EHR data extraction) are defined as post-COVID measures”. Could the interval of time between COVID-19 diagnosis and CGI-S scale affect the results?

> This is an interesting question, which could be addressed in a future study that has sufficient number of patients who are assessed at specific post-COVID time intervals. 

RESULTS.

Please report data as mean and standard deviation or median and interquartile range.

> We updated the Table and on page 7 we wrote: “A total of 56 patients (mean age 76.0 ± 8.5 years; median age: 74.5 years; interquartile range: 13.0 years) (Table; Table S2) with confirmed COVID-19 were reported to the treating psychiatrists between February and September 2020.”

DISCUSSION.

“As older adults are particularly vulnerable to the negative consequence of antipsychotic exposure and to higher mortality risk from COVID-19”. Please add a reference.

> References were added as requested.

6. PLOS authors have the option to publish the peer review history of their article (what does this mean?). If published, this will include your full peer review and any attached files.

>No.

Do you want your identity to be public for this peer review? For information about this choice, including consent withdrawal, please see our Privacy Policy.

Reviewer #1: No

Reviewer #2: No

---

## [Editor Report · Decision Letter 1]

20 Sep 2021

PONE-D-21-17894R1Mortality and psychiatric outcome among geriatric psychiatry outpatients with Coronavirus disease (COVID-19)PLOS ONE

Dear Dr. Freudenberg-Hua,

Thank you for submitting your manuscript to PLOS ONE. After careful consideration, we feel that it has merit but does not fully meet PLOS ONE’s publication criteria as it currently stands. Therefore, we invite you to submit a revised version of the manuscript that addresses the points raised during the review process.

We look forward to receiving your revised manuscript.

Kind regards,

Ismaeel Yunusa, PharmD, PhD

Academic Editor

PLOS ONE

Journal Requirements:

Additional Editor Comments:

Thank you for the revision. Can you please modify your title to: “Mortality and clinical outcomes in association with antipsychotic medication use among geriatric psychiatry outpatients with COVID-19"
---

## [Author Response · Author response to Decision Letter 1]

24 Sep 2021

No reviewers' comments were presented for this minor revision. We modified the title to "Mortality in association with antipsychotic medication use and clinical outcomes among geriatric psychiatry outpatients with COVID-19" in response to the editor's suggestion.

---

## [Editor Report · Decision Letter 2]

8 Oct 2021

Mortality in association with antipsychotic medication use and clinical outcomes among geriatric psychiatry outpatients with COVID-19

PONE-D-21-17894R2

Dear Dr. Freudenberg-Hua,

We’re pleased to inform you that your manuscript has been judged scientifically suitable for publication and will be formally accepted for publication once it meets all outstanding technical requirements.

Kind regards,

Ismaeel Yunusa, PharmD, PhD

Academic Editor

PLOS ONE
---

## [Editor Report · Acceptance letter]

13 Oct 2021

PONE-D-21-17894R2 

Mortality in association with antipsychotic medication use and clinical outcomes among geriatric psychiatry outpatients with COVID-19 

Dear Dr. Freudenberg-Hua:

I'm pleased to inform you that your manuscript has been deemed suitable for publication in PLOS ONE. Congratulations! Your manuscript is now with our production department. 

Kind regards, 

on behalf of

Dr. Ismaeel Yunusa 

Academic Editor

PLOS ONE